# The Zebrafish, an Outstanding Model for Biomedical Research in the Field of Melatonin and Human Diseases

**DOI:** 10.3390/ijms23137438

**Published:** 2022-07-04

**Authors:** Paula Aranda-Martínez, José Fernández-Martínez, Yolanda Ramírez-Casas, Ana Guerra-Librero, César Rodríguez-Santana, Germaine Escames, Darío Acuña-Castroviejo

**Affiliations:** 1Biomedical Research Center, Department of Physiology, Faculty of Medicine, Institute of Biotechnology, Technological Park of Health Sciences, University of Granada, 18016 Granada, Spain; ampaula@correo.ugr.es (P.A.-M.); josefermar@ugr.es (J.F.-M.); yolandaramirez@correo.ugr.es (Y.R.-C.); aguerit@ugr.es (A.G.-L.); cesar@correo.ugr.es (C.R.-S.); gescames@ugr.es (G.E.); 2CIBERfes, Ibs.Granada, 18016 Granada, Spain; 3UGC of Clinical Laboratories, San Cecilio Clinical University Hospital, 18016 Granada, Spain

**Keywords:** zebrafish, melatonin, clock genes, pathology

## Abstract

The zebrafish has become an excellent model for the study of human diseases because it offers many advantages over other vertebrate animal models. The pineal gland, as well as the biological clock and circadian rhythms, are highly conserved in zebrafish, and melatonin is produced in the pineal gland and in most organs and tissues of the body. Zebrafish have several copies of the clock genes and of *aanat* and *asmt* genes, the latter involved in melatonin synthesis. As in mammals, melatonin can act through its membrane receptors, as with zebrafish, and through mechanisms that are independent of receptors. Pineal melatonin regulates peripheral clocks and the circadian rhythms of the body, such as the sleep/wake rhythm, among others. Extrapineal melatonin functions include antioxidant activity, inducing the endogenous antioxidants enzymes, scavenging activity, removing free radicals, anti-inflammatory activity through the regulation of the NF-κB/NLRP3 inflammasome pathway, and a homeostatic role in mitochondria. In this review, we introduce the utility of zebrafish to analyze the mechanisms of action of melatonin. The data here presented showed that the zebrafish is a useful model to study human diseases and that melatonin exerts beneficial effects on many pathophysiological processes involved in these diseases.

## 1. Zebrafish as a Model for Biomedical Research

### 1.1. General Features of Zebrafish

*Danio rerio*, usually called the zebrafish, is a vertebrate of the Cyprinidae family, belonging to teleost and freshwater fish, whose origin comes from Southeast Asia [1]. It takes its name from the horizontal blue-pigmented stripes that go from the operculum to the tail giving it the zebra appearance [2]. The zebrafish is an adult and sexually matures at 3 months of age. Adults are 4–5 cm in length and can live up to 5 years. This species is sexually dimorphic; males have a golden color and are smaller than females because the latter are full of eggs and silvered [1].

Zebrafish can reproduce throughout the year and the embryos develop rapidly [1]. The embryos begin as a single-cell bud that divides into thousands of cells. The cells migrate down the sides of the bud to form the head and tail which will separate from the body. At 36 h after fertilization (hpf), the precursors of all organs appear except for cardiac physiology, which is not stable until 4 days post fertilization (dpf). From 5 dpf, they respond to optical, tactile, and acoustic stimuli, show free swimming, and the digestive and excretory system are fully functional [3].

### 1.2. Advantages of Using Zebrafish as an Experimental Model

Zebrafish were first introduced to biomedical research by George Streisinger in 1981 [4] and they are currently considered a suitable model to investigate development, genetics, immunity, behavior, physiology, nutrition, neurodegeneration, clock and circadian rhythms, among others, and it is considered an excellent model for human diseases [5]. This is because they offer many advantages over other vertebrate animal models: small size, which makes them easier to manipulate; a rapid embryonic development with most of the organs fully formed at 2 dpf; the transparency of the embryos that allows to observe them at different stages of the development; and their high reproductive capacity. Another advantage over mammals is that zebrafish embryo development is extrauterine, which makes it easy to study and observe the early stages of development, making it an excellent model for the study of developmental biology. These advantages also facilitate the early manipulation of zebrafish larvae through targeted genetic manipulation techniques such as transgenesis, TALEN, CRISPR, RNA overexpression, and others [6,7,8]. In addition, the use of zebrafish is much cheaper than small mammals including mice and rats, because the maintenance of the fish colony comes at a relatively low cost. Moreover, chemicals are directly administered into the water and are absorbed through the skin of the fish, leading to less discomfort than mammals that are injected with these chemicals. In this connection, the zebrafish is a useful model for drug screening focused for example on organ toxicity [9], development [9], respiration [10], etc.

Thanks to the transparency of the larvae, it is possible to monitor cells or structures in vivo, through the development of transgenic lines marked by fluorescence [11]. Another feature is to perform confocal microscopy on the complete larva, without making sections. Furthermore, the zebrafish genome is completely sequenced and there is a great similarity between the zebrafish and the human genome. They share 71% of genes and 82% of the genes involved in some diseases [12]. They also present a high physiological similarity in genetic mechanisms of development and the process of aging and death. Although it is true that embryonic development in zebrafish is extrauterine, it has orthologs of many human genes such as *hes5*, *sox11*, *dlx2*, *ihh*, and *gata2*, among others involved in embryogenesis [13]. Finally, an outstanding feature of zebrafish embryos is the possibility to measure mitochondrial respiration in vivo [10], which allows a better approximation of the effects of a disease and/or treatments on the bioenergetic capacity of them.

For all these reasons, the zebrafish has become a good model for the study of human diseases. To date, the zebrafish has been used as an experimental model for the study of neurological disorders, genetic disorders, cardiovascular diseases, and others [14].

### 1.3. Circadian Rhythms in Zebrafish

The significance of zebrafish for circadian clock studies depends on the ease to carry out genetic experiments through which mutant genes and their functions can be identified, whereas the transparency of the embryos allows to obtain neuronal images in vivo. The biological clock of zebrafish, similar to other vertebrates, has a rhythm period of approximately 24 h, and is made up of a feedback loop. The clock consists of two activating elements, BMAL1 and CLOCK, that form a heterodimer and activate the transcription of the inhibitors PER and CRY, and the modulators ROR-α y REV-erb. PER and CRY dimerize and translocate to the nucleus to inhibit the transcriptional activity of BMAL1/CLOCK [15,16]. ROR-α enhances the expression of BMAL1 whereas REV-erb α inhibits it (Figure 1).

This molecular loop is more complex in zebrafish than other vertebrates because the former contains several copies of the key clock genes. Three *bmal* genes (*bmal1a*, *1b* and *2*) and three *clock* genes (*clock 1a*, *1b* and *2*) have been identified and they form heterodimers by different combinations [17,18]. These combinations of heterodimers present differences in the efficiency of gene transcription activation and in the susceptibility to being inhibited. Unlike mammals, where *clock* has not rhythmic expression, both *bmal* and *clock* gene families show a rhythm of expression in zebrafish that reaches its maximum peak after the transition from light to dark [16]. Regarding *cry*, six genes have been observed: *cry1aa*, *1ab*, *1ba*, *1bb*, *2* and *3*, and they are divided into two types. Cry1aa, cry1ab, cry1ba and cry1bb share sequence homology and function with mammalian *cry*, and so they inhibit the transcriptional activation of CLOCK: BMAL1 and are called Repressive Cry (RC) [19]. The others, *cry2* and *cry3*, are called Non-Repressive Cry (NRC), do not repress *clock* and *bmal* activation [19], and its function has not been determined yet. In addition, the rhythmic expression of *cry* genes varies as *cry1aa* is light-driven, whereas the other *cry* genes are clock-regulated [20]. Furthermore, *cry2* has its maximum expression peak in the morning; *cry1aa*, *1ab*, and *3* during the mid-light phase, and *cry1ba* and *1bb* at night. Finally, there are two homologues of *per1*, *per1a* and *1b*, in addition to *per2* and *per3* [21]. Here, *per2* is light-driven, whereas the remaining *per* genes are clock-regulated. *Per2* has been shown to work in conjunction with *cry1* [21].

The interest of zebrafish to study circadian rhythms is increasing because its physiological and behavioral rhythms develop very early and rapidly in this animal, settling at 4–5 dpf. Furthermore, zebrafish present a highly conserved physiological rhythms in addition to the rhythm of melatonin production [22]. During embryonic development, it has been shown that the cell cycle is under the control of light because the cells enter the S phase of the cell cycle at the end of the day, when the larvae are exposed to the light–dark rhythm [23]. Moreover, at 4 dpf, it can be detect a stable diurnal rhythm of locomotor activity [24]. In adults, diurnal feeding is also controlled by the circadian system through the glucocorticoid receptor [25]. In addition, cortisol, which also regulates the response to stress, influences behavior, growth, reproduction, and osmoregulation [26]. The control of these and other rhythms in zebrafish by melatonin will be discussed.

## 2. Melatonin in Zebrafish

The central clock in mammals is in the suprachiasmatic nucleus (SCN) and drives the photic inputs to the pineal gland to produce melatonin secretion. A similar nucleus in zebrafish has been not identified, but the pineal gland works as the central pacemaker because it contains photoreceptive cells and contains a full circadian oscillator. The zebrafish is a diurnal animal with a nocturnal production of melatonin.

The pineal gland and the sleep regulatory mechanisms are highly conserved in zebrafish. The pineal gland of the zebrafish is developed very early and it contains photoreceptive cells similar to mammalian retinal photoreceptors [27]. Unlike in mammals, who have a clock structure in the suprachiasmatic nucleus (SCN) of the brain, the pineal photoreceptor cells work as the intrinsic circadian clock [28]. As in mammals, this intrinsic pineal circadian clock drives the rhythmic synthesis of melatonin, being highest at night and low during the day. Melatonin synthesis is regulated by aralkylamine N-acetyltransferase (AANAT) and zebrafish, like other teleosts, have two *aanat* genes [29,30]. Therefore, in zebrafish, the pineal gland is the central pacemaker that receives information from ambient light and translates it into a neuroendocrine signal, i.e., melatonin production [31]. Nevertheless, recent results showed that most zebrafish tissues contain molecular clocks that are also directly influenced by light, suggesting that the pineal gland modulates behavioral circadian rhythms in a multiple pacemaker system [32]. It is then plausible that the pineal zebrafish acts as the central clock that drives the synchronization of the other “peripheral clocks” resembling the control of the SCN master clock in mammals.

### 2.1. Synthesis and Metabolism of Melatonin

Melatonin, or N-acetyl-5-methoxytryptamine (aMT), is an indolamine synthesized from tryptophan and it is highly conserved in zebrafish [33]. Dermatologist Aaron Lerner in 1958–1959 [34] isolated and identified a product from the beef pineal gland that lightens skin color in amphibians [35], and inhibits melanocyte-stimulating hormones (MSH), which was termed melatonin. This pineal melatonin was deeply studied as the factor that controls the circadian rhythms. Today, however, we know that melatonin is produced in most of the organs and tissues of the body; it has different properties than the pineal melatonin and is called extrapineal melatonin [36].

As in mammals, the zebrafish exhibits maximal melatonin production at night, generated by the SCN clock in the former and associated with the light–dark period [37]. Extrapineal melatonin, however, does not have a circadian rhythm in mammals and it is found at much more higher concentrations than the pineal one [36,38].

The synthesis pathways of mammalian pineal and extrapineal melatonin are quite similar, and it is thought that the zebrafish shares equivalent ways. Tryptophan is taken up from the circulation and hydroxylated by L-tryptophan hydroxylase (TPH) at position 5, generating 5-hydroxytryptophan, which is then decarboxylated to form serotonin (5-hydroxytryptamine) by 5-hydroxytryptophan decarboxylase. Serotonin is acetylated by AANAT, originating N-acetylserotonin (NAS) that is methylated by N-acetylserotonin O-methyltransferase (ASMT) producing melatonin (Figure 2). Once synthetized, and because it is not stored in the gland, pineal melatonin quickly diffuses into the systemic and cerebral bloodstream, being distributed throughout the body. By contrast, extrapineal melatonin remains into the cell and it does not go to the extracellular fluid [37]. The key melatonin synthesis enzymes, AANAT and ASMT, are expressed in almost all body tissues [39], but only in the pineal (and in the retina) does AANAT exhibit the same circadian rhythm as melatonin [40].

As mentioned, zebrafish have two *aanat* genes because of duplications of the genome, and they show different distributions. Whereas *anaat1* is expressed only in the retina and it depends on the photoperiod, *aanat2* is expressed in the pineal gland and, at lower levels, in the retina and other tissues (18), and is controlled by the circadian clock. Therefore, zebrafish shows both pineal and extrapineal melatonin synthesis.

Most of the circulating melatonin is bound to albumin and a small percentage (30%) is free, passing into the saliva in mammals. In addition, its half-life in plasma is 30 min, being metabolized by cytochrome P450 (CYP) in the liver producing 6-hydroxymelatonin [33,41]. 6-hydroxymelatonin is conjugated with sulfate and 6-sulfatexymelatonin is formed and excreted in the urine [41,42]. On the other hand, extrapineal melatonin is metabolized in situ, i.e., yielding N^1^-acetyl-N^2^ formyl-5-methoxy-kinuramine (AFMK) by cleavage of the pyrrole ring of melatonin [43]. This reaction is produced by arylamine formamidase, or through the interaction of melatonin with a reactive oxygen species (ROS). A subsequent reaction of AFMK with ROS yields N^1^-acetyl-5-methoxyquinuramine (AMK) [43]. Both metabolites of melatonin, AFMK and AMK, share antioxidant and anti-inflammatory properties with melatonin [44,45]. Pineal and extrapineal melatonin metabolism is not yet known in zebrafish but it should occur as it does in mammals, because AFMK and AMK have been detected in zebrafish embryos [10].

### 2.2. Mechanisms of Action of Melatonin

Multiple actions of melatonin have been described: mediated by specific receptors on cell membranes [46], mediated through nuclear receptors [36,47], and mediated but acting at the intracellular level by binding to cytosolic proteins [36,48,49]. Moreover, some of the actions of melatonin are independent of receptors, including its ability as a free radical scavenger [50]. Therefore, the mechanisms of action of melatonin can be classified as receptor-mediated and receptor-independent ones [51]. Thus, melatonin has multiple targets including membrane and nuclear receptors.

Melatonin membrane receptors are G protein-coupled receptors and are divided into three main subtypes, MT1 (or Mel1A or MTNR1A) and MT2 (or Mel1b or MTNR1B), and Mel1c. Mel1a and Mel1b were identified in almost all vertebrate species, whereas Mel1c was only found in non-mammalian vertebrates. As for other genes, zebrafish have duplicated mtnr1a in to mtnr1aa and mtrn1ab, and mtnr2 into mtnr2bb and mtnr2ba [52]. All of these receptors are present in zebrafish and expressed in some conditions [53]. These receptors are more expressed during developing that in adult zebrafish, and they are related to accelerating cell proliferation and development of zebrafish embryos [54]. Recently, it has been shown that teleosts have another subtype, the Mel1a-like gene, also called Mel1D. In mammals, the latter is missing and Mel1c has lost the ability to bind melatonin [52,55].

In mammals, the activation of Mel1a and Mel1b receptors cause the dissociation of G protein into the α and βγ dimers that interact with various effector molecules [56]. When melatonin binds to the Mel1a receptor, adenylate cyclase is inhibited and cAMP production decreases. This impedes CREB phosphorylation since protein kinase A is inhibited and intracellular calcium increases due to the activation of protein kinase C. Binding to the Mel1b receptor also inhibits the guanylate cyclase pathway, thus decreasing cGMP (Figure 3) [57].

On the other hand, orphan retinoid-related receptors (ROR) are a family of nuclear hormone receptors consisting of RORA, B, and C, more commonly known as ROR α, β, and γ, respectively. RORs bind to DNA ROR response elements (RORE) thanks to their N-terminal domain and through which they regulate gene transcription (Figure 3). In their C-terminal region, they bind to their ligands including melatonin to RORα ligand [58]. Orthologs for the ROR gene family have been cloned and their participation in embryo development is also supported [58]. Additional effects of the ROR receptors family include anti-inflammatory actions [59]. Specifically, melatonin binds to ROR α, activating SIRT1 that in turn deacetylates NF-κB, inhibiting its bind to DNA in mammals [60], although it is unclear whether this effect is also present in zebrafish.

Melatonin nuclear and membrane receptors are known to exhibit circadian rhythms in rat livers. The acrophases of mRNA expression took place at 03:00 h, one hour after the melatonin acrophases, so these changes seem to be regulated by the levels of melatonin in the blood [61]. In zebrafish, melatonin receptors also change in the function of the light– dark environment and other factors that influence them [53,62].

Moreover, it has been shown in mammals that the effects of melatonin can also be independent of membrane receptors and interact with cytosolic proteins. Some cytosolic proteins to which melatonin binds are the calmodulin [63], calreticulin [49], tubulin [64], and protein kinase C (Figure 3) [65], all of which are involved in calcium metabolism and modulation of the cytoskeleton structure [66].

The melatonin receptor-dependent and independent effects were recently demonstrated in our lab in a zebrafish model of Parkinson’s disease. Mel1a and Mel1b receptors were blocked with luzindole Mel1a+Mel1b antagonist) and 4PPDOT (Mel1b antagonist). We showed that melatonin, even blocking its membrane receptors, acts as an antioxidant, increasing the expression of antioxidant enzymes and inhibiting prooxidant enzymes when is administered to MPTP-treated zebrafish embryos (data not published yet).

### 2.3. Pineal Melatonin Properties

Pineal melatonin released into the circulation reaches every cell in the body, where it acts as a chronobiological signal controlling the peripheral clocks of different tissues. Here, melatonin coordinately controls the basic chronobiotic functions of cells and organs, including circadian and seasonal adaptative changes [67].

The first rhythm controlled by melatonin is cell proliferation in the zebrafish embryo and, therefore, its development. This action is believed to be mediated by the membrane receptor Mel1b and cell proliferation in a tissue is determined by the Mel1a/Mel1b ratio. In addition, Mel1b is found in greater proportion in the zebrafish embryo and Mel1a in adult fish [54]. Moreover, the rate of cell proliferation is higher at night, coinciding with the secretion of melatonin, which is thought to be an adaptive mechanism to avoid the damage that UV radiation can cause to DNA during the day [23]. Lastly, both light and melatonin are thought to be necessary for neuronal differentiation and proper growth of dendrites in zebrafish [68].

The sleep/wake rhythm is one of the best known and studied melatonin-controlled rhythms in zebrafish since it is a diurnal species and presents similarities in the behavioral patterns of sleep in mammals [69]. At 4–5 dpf and until adulthood, zebrafish exhibit daily fluctuations in locomotor activity, exhibiting maximal activity during the light phase and inactivity during the dark phase with reduced sensory response. This sleep behavior is regulated by the circadian system through melatonin acting on its membrane receptors [70]. With age, there is a decrease in melatonin production, but the expression of its membrane receptors is not altered, so melatonin administration continues to promote sleep in aged zebrafish [71]. Recently, the relationship between hypocretin with melatonin to regulate the sleep circadian rhythm in zebrafish has been proposed, and it was suggested that the hypocretin signaling pathway regulates the expression of *aanat2* and, consequently, the production of melatonin. Furthermore, the spatial distribution of hypocretin receptors closely resembles the expression pattern of melatonin receptors [72].

At 5 dpf, the fish also begin to feed, melatonin being part of the signal network that regulates appetite in zebrafish, stimulating anorexigenic signals and inhibiting orexigenic ones. Chronic administration of melatonin in water decreases food intake, thus controlling the balance between energy intake and expenditure. There is an increase in leptin and a decrease in ghrelin levels in the brain after melatonin administration [73]. In addition, it has also been observed that melatonin influences the endocannabinoid system, through the regulation of CB1 expression [73]. More recently, it was shown that the increased expression of leptin, thanks to melatonin, also occurred in the liver and intestine as well as in the brain [74]. However, no melatonin receptors were found in the gut and liver of zebrafish, suggesting that the effect of melatonin is brain-mediated [74].

The zebrafish, as already mentioned, is a diurnal species and it has a reproductive rhythm, in which light is a key factor to regulate the time of day in which the zebrafish spawns, which occurs at dawn [75]. Moreover, Blanco-Vives et al. [75] showed that one-hour darkness disruption of the normal light–dark cycle affected spawning rate, although feeding time did not disrupt it, thus emphasizing that the reproduction rhythm in zebrafish is controlled exclusively by light. Melatonin regulates the endocrine factors involved in growth and maturation of oocytes, increasing the fertility of zebrafish. These effects of melatonin are not produced by influencing gonadotropin-releasing hormone (GnRH) directly but by acting on kisspeptins, kiss1, and kiss2 [76]. Activation of kiss1 and kiss2 gene transcription in the zebrafish brain by melatonin stimulates GnRH production, increasing fertility [76]. In addition, the zebrafish gonad shows a daily rhythm of the hormones involved in oocyte development, such as activin βA, activin βB, and follistatin. Activin βA and follistatin peak during the dark phase and are responsible for ovarian and follicle growth. However, activin βB, which regulates the last stage of oocyte maturation and ovulation, reaches its highest level in the hour before lights are turned on, which could explain spawning at dawn [77]. Additionally, follicle-stimulating hormone (FSH) and luteinizing hormone (LH) have daily activity that influences the last stages of follicular development, increasing 6 h before the onset of ovulation [78]. An increase in the vitellogenin and gonadosomatic index was found in zebrafish livers after melatonin administration, suggesting its positive role in kiss1 and kiss2-dependent oocyte maturation [76]. Moreover, an increase in the signals involved in the final maturation of the oocyte in the follicles with melatonin was also observed, suggesting that melatonin also has local action. This idea has been supported by the discovery of Mel1a receptors in oocytes [76].

All these melatonin-controlled rhythms are influenced by the photoperiod, but it has been shown that temperature also modulates melatonin synthesis through the enzyme AANAT2 [79]. Thus, melatonin could be responsible for entraining the rhythm of expression of clock genes due to temperature changes. Lahiri et al. have observed that in zebrafish larvae exposed to total darkness (DD), the temperature established the circadian rhythm of the clock genes, except *per2* because is light-driven [80]. In addition, they demonstrated that temperature changes lead to changes in the rate of expression and transcription of clock genes, with the transcriptional activity of *clock* and *bmal1* being temperature-dependent [80].

### 2.4. Extrapineal Melatonin Properties

Whereas pineal melatonin has chronobiotic functions, extrapineal melatonin acts as an antioxidant and anti-inflammatory, and its main target of action is the mitochondria.

The antioxidant activity of melatonin depends on its ability to be produced in the cells of organs and tissues of the body [81]. Both melatonin and its metabolites, AFMK and AMK, are capable of scavenging free radicals directly [82], interacting with ROS including hydroxyl (OH•) and superoxide radical (O_2_^−^•). Melatonin can also interacts with peroxynitrites (ONOO^−^) and scavenges peroxyl radicals (LOO•) [83]. Melatonin is capable of lowering malondialdehyde levels, a marker of oxidative stress, supporting the role of the former in scavenging ROS in this animal species [84].

It has been shown in zebrafish, as well in mammals, that melatonin is also capable of indirectly scavenging free radicals by regulating the activity and expression of other antioxidant systems, an effect that could be mediated by its nuclear receptor [10]. Firstly, melatonin increases the activity of glutathione peroxidase (GPx) and glutathione reductase (GRd), stimulating the glutathione cycle [85,86] and, thus maintaining the balance of oxidized glutathione/reduced glutathione (GSSG/GSH). In addition, it stimulates γ-glutamylcysteine synthase, increasing the production of GSH [87] and glucose-6-phosphate dehydrogenase (G6PD), which generates the NADPH required by GRd [88]. Finally, melatonin also enhances the activity and expression of other antioxidant enzymes such as superoxide dismutase (SOD) and catalase [81,89].

Therefore, melatonin is a powerful antioxidant that regulates the redox state of the cell and because mitochondria is the major source of free radicals, these organelles are its main target of action [38,90]. Its action in the mitochondria is not only due to its antioxidant role, but it also plays a fundamental task in the maintenance of mitochondrial homeostasis, since it preserves the stability, integrity, and function of the mitochondrial membranes [91,92]. In zebrafish, melatonin also acts directly on the electron transport chain (ETC), where it increases the activity of the four respiratory complexes I–IV [10] (Figure 3).

Melatonin can also act as an anti-inflammatory in mammals, where it is known to regulate the NF-κB/NLRP3 inflammasome pathways, modulating the expression of proinflammatory cytokines, tumor necrosis factor alpha (TNF-α), and inducible nitric oxide synthesis (iNOS), among others [60,93]. Thus, melatonin prevents the activation of the NLRP3 inflammasome and the formation of caspase 1 as well as IL-1β (Figure 3) [93]. Furthermore, melatonin inhibits the expression of cyclooxygenase (COX), preventing the excessive production of inflammatory mediators [94].

Little is known about the anti-inflammatory action of melatonin in zebrafish. If it has been observed that it inhibits the expression and activity of iNOS [95]. In addition, as in mammals, melatonin regulates the recruitment of neutrophils and adhesion molecules to the site of injury through the regulation of proinflammatory cytokines such as IL-8 and TNF-α [96,97]. Finally, melatonin also plays an important role in apoptosis (Figure 3), both in mammals and fish, by inhibiting the proapoptotic factor Bax and the caspase pathway, and activating Bcl-2 [84].

## 3. Therapeutic Properties of Melatonin in Zebrafish Models of Disease

### 3.1. Melatonin in Parkinsonian Zebrafish

Parkinson’s disease is a neurodegenerative disorder that affects the area of the central nervous system responsible for controlling the motor system, called substantia nigra. Parkinson’s is characterized by the progressive loss of dopaminergic neurons and the consequent lack of dopamine [98].

There are two forms of Parkinson’s disease, familial and idiopathic. The hereditary form occurs in approximately 5–10% of Parkinson’s patients due to mutations in the *park* genes, which code for proteins responsible for mitochondrial homeostasis. Mutations in α-synuclein form protein aggregates in dopaminergic neurons called Lewy bodies that appear during this disease in humans [99]. Zebrafish have the γ-synuclein which behaves in the same way, essential for spontaneous movement and dopaminergic functions [100]. Another of these mutations occurs for the *parkin* and *pink1* genes, also present in zebrafish [101]. PARKIN is phosphorylated and activated by PINK1, and these initiate the autophagy of damaged mitochondria [102]. Finally, mutations in *dj-1*, which regulate the expression of antioxidant genes [103], and in tyrosine hydroxylase (TH) [104], were reported in zebrafish also. TH is an enzyme that catalyzes the transformation of L-dihydroxyphenylalanine or L-DOPA into dopamine, so it is another marker of Parkinson’s disease. However, most cases of Parkinson’s (between 90–95%) are of unknown cause, although the main factor is aging.

The alterations in mitochondrial function produce an increase in oxidative stress [105] and, together with neuroinflammation, are main processes that occur in any form of Parkinson’s disease, leading to dopaminergic cell death [106]. During Parkinson’s disease, the inhibition of the complex I occurs, favoring the formation of ROS and reactive nitrogen species (RNS), which further inhibit the ETC complexes and produce more oxidative damage to the mitochondria [82]. These events lead to a decrease in mitochondrial inner membrane potential, reducing the ATP production, and the opening of the mitochondrial permeability transition pore (mTP) that releases cytochrome c to the cytosol to start the apoptosis cascade. Moreover, overactivation of the microglia in the substantia nigra collaborates with neuroinflammation and subsequent neuronal damage and furthermore, microglia activation [107].

At 72 hpf, the catecholaminergic system in zebrafish is already developed and is equivalent to dopaminergic neurons in mammals [108]. In this regard, zebrafish embryos are sensitive to MPTP (1-methyl-4-phenyl-1,2,3,6-tetrahydropyridine), a neurotoxin that induces a parkinsonian state such as in mammals by dopamine depletion because the former use the dopamine uptake mechanism to enter the dopaminergic cell, preventing the re-uptake of dopamine that is oxidized at extracellular level inducing a huge amount of ROS. Once into the cell, MPTP inhibits complex I, yielding an oxidative stress and mitochondrial damage terminating in neuronal death [109].

The role of melatonin in Parkinson’s disease has been also studied in zebrafish treated with MPTP. This model of parkinsonism in zebrafish embryos has been recently published [95] and, besides other results, it was showed for the first time the changes in mitochondrial function in vivo after MPTP and/or melatonin administration.

Melatonin administration to MPTP-induced parkinsonism in zebrafish embryos counteracts oxidative stress increasing the expression and activity of antioxidant enzymes and maintaining the GSSG/GSH ratio [10]. In addition, melatonin restored the mitochondrial bioenergetics affected by MPTP, restoring the ATP production [10]. In addition, melatonin counteracted the reduction in autophagy and the increase in mitochondrial fusion induced by MPTP [10]. Moreover, the effect of melatonin in parkinsonian-related gene expression was analyzed in the same model. Here, melatonin increased the expression of *parkin*/*pink1*/*dj-1* genes that were decreased by MPTP, being able to carry out mitophagy of damaged mitochondria. Phenotypic alterations, including motor ones, in parkinsonian zebrafish embryos were also counteracted by melatonin. All these features of parkinsonism were prevented and recovered by melatonin administration. For the first time, and besides being preventive, curative properties of melatonin were observed for restoring normal complex I activity and reducing the expression of γ-synuclein and iNOS, TH [95].

### 3.2. Melatonin against Epilepsy in Zebrafish

Epilepsy is a chronic neurological disorder that affects 1–2% of the world’s population and is considered potentially untreatable [110]. The signs of the disease are presented as repeated spontaneous seizures and abnormal neuronal discharges in brain regions [111]. In addition to the characteristics of this pathology, 30–50% of patients suffer from epileptic comorbidities such as depression and anxiety [112]. Unfortunately, existing treatments may improve seizure frequency but do not reduce epileptic comorbidities, and may even worsen them [113].

Lower plasma melatonin levels have been observed more in epileptic patients than in healthy controls [114]. Moreover, excitatory neurotransmitters such as glutamate and aspartate are involved in the generation of seizures, and melatonin was reported to inhibit glutamatergic activity in rat brain [115]. The effects of melatonin include the inhibition of the mediator of the NMDA glutamatergic receptor activation, nNOS, thus reducing NO and excitotoxic response [116]. In a model of generalized epilepsy in rats treated with pentylenetetrazol (PTZ), melatonin reduced seizures affecting brain amino acids and NO levels [117,118].

For all these reasons, Qingyu Ren et al. have studied the role of melatonin in epileptic comorbidities using zebrafish as a model. To do this, they treated zebrafish embryos at 6 dpf with melatonin and PTZ at 7 dpf. They found that melatonin reversed the decreased expression of genes implicated in depression and anxiety to control levels. On the other hand, since seizures are caused by the excitotoxicity of glutamate, which generates oxidative stress, they analyzed the expression of antioxidant enzymes and observed that they increased after melatonin treatment. Even more, the prooxidant enzymes that were elevated increased in the PTZ group were reversed by melatonin. This suggests that melatonin improves seizures and comorbidities through glutathione metabolism [119].

### 3.3. Protective Effects of Melatonin against Cardiovascular Damage in Zebrafish

The zebrafish is being used for the study of the cardiovascular system in vertebrates because cardiac function, the development of the heart and blood vessels, and the features of heart disease are similar tohumans [120]. Melatonin is proposed as a treatment for cardiovascular diseases due to its role in the regulation of cell proliferation and inflammation that protects the cardiovascular system. In addition, the alteration of the circadian rhythm increases the risk of these diseases [121].

On the one hand, the pesticide deltamethrin (DM) has been used in zebrafish larvae to generate a model of pericardial edema that leads to decreased cardiac function. Melatonin restored heart rate, ventricular stroke volume, and cardiac output to the control level. It also increased the expression of sodium channel genes affected by DM [122]. Moreover, the alteration of the circadian system and thyroid hormones by addition of 6-benzylaminopurine (6-BA) to zebrafish embryos induces cardiac morphological abnormalities including pericardial edema, linearization of the heart, vascular development defects, and slow heartbeat. After restoring the circadian function after melatonin administration in 6-BA-treated zebrafish, thyroid hormones were normalized and morphology and functions of the cardiac system, restored [123]. These data speak about the utility of melatonin in cardiovascular diseases.

### 3.4. Melatonin and Obesity in Zebrafish

The number of people with obesity is increasing exponentially and represents the fifth cause of mortality worldwide due to the development of risk factors such as diabetes, cardiovascular diseases, and cancer, leading to premature death [124]. The cause of fat gain depends on disturbances in the complex network that controls balance between caloric intake and energy expenditure [125].

The circadian activity of melatonin regulates a series of rhythms, including food intake and metabolism, adiposity, and seasonal variation of body weight, glucose uptake and gut reflexes [126].

The zebrafish is used as a model for metabolic diseases because it can mimic diet-induced obesity, accumulating fat just as in humans [127]. In obese zebrafish model, melatonin treatment reduced body mass index (BMI) and weight, through a mechanism involving carbohydrate and unsaturated fatty acid metabolism. Melatonin also reduces the size of adipocytes, allowing their participation in metabolic processes, and inhibits white adipogenesis, reducing body weight. Moreover, melatonin controls the expression of genes related to appetite and satiety control, such as leptin and ghrelin, both in the brain and in the intestine. Food intake, however, was unaffected by melatonin treatment, meaning that melatonin regulates energy homeostasis in zebrafish independently of food intake. Therefore, treatment with melatonin is suitable for obesity control and loss of weight gain [128]. Nevertheless, additional studies on melatonin action in obese zebrafish are necessary to know the molecular pathways involved in its effects to see whether it may be applicable to humans.

### 3.5. Regulation of Bone Metabolism by Melatonin in Zebrafish

There is evidence showing immune activity in a bone fracture, releasing cytokines that stimulate bone repair by macrophages, neutrophils, and lymphocytes [129]. Given the action of melatonin on the immune system, its relationship with bone repair has been studied [130,131,132]. It has been observed that in vitro melatonin promotes bone proliferation by stimulating osteoblast proliferation [133] and inhibits bone resorption through the inhibition of the receptor activator of nuclear factor κ B (RANK) and, therefore osteoclast differentiation [134]. Conversely, in an in vivo mouse femur fracture model, melatonin was shown to hinder bone healing [135]. Thus, there is controversy about the functions of melatonin on bone repair and, as aconsequence, the zebrafish emerges as a model for these studies.

Zebrafish scale is a bone tissue, similar to the membranous bone tissues of mammals, and undergoes continuous metabolism throughout its life [136]. Furthermore, osteoblasts and osteoclasts are highly conserved both in zebrafish and mammals [137]. Kobayashi-Sun et al. [138] reported that melatonin inhibits the differentiation of both osteoblasts and osteoclasts in the zebrafish scale because it prevents the Erk signaling cascade. These results contrast with those obtained in the in vitro model of mammalian cells, and also in vivo in which melatonin increases bone formation [139]. This negative effect of melatonin on osteoblasts has also been seen when incubated with osteoclasts [140]. However, it is known that with age there is a reduction in melatonin that accompanied osteoporosis and bone loss [133], and melatonin therapy may repair bone fractures because it enhances osteoblastogenesis [141]. To further better understand the role of melatonin in bone formation and repair, further experiments are necessary, and the zebrafish becomes in an excellent model to study fracture healing thanks to the in vivo imaging of osteoblasts and osteoclasts on the scales.

### 3.6. Melatonin in Zebrafish Model of Psychiatric Diseases

The zebrafish is an excellent model for research in pharmacology, due to its potential for the detection of central nervous system drugs. The use of melatonin as an anxiolytic was studied in preoperative anxiety and analyzed for its post-operative effect [142,143]. Melatonin has a hypnotic and sedative effect because it favors the binding of γ-aminobutyric acid (GABA_A_) to its receptor, in the same way that benzodiazepines, without also having no effect on cognitive and motor function [144,145,146]. The benzodiazepine-like effects of melatonin are counteracted by naloxone, suggesting the participation of opioid peptides in these effects [147].

Adult female zebrafish exhibit greater anxiety-like behaviors [148], activity [149], and endocrine (cortisol) responses [150] than males. This is possibly due to lower levels of testosterone and higher levels of estrogen. Anxiety was assessed with the novel tank test, using the parameters of freezing time and freezing episodes as anxiety behaviors, and time on top of the tank as anxiolytic behavior. Both males and females respond equally to the anxiolytic effect of melatonin. Here, less time and freezing episodes were obtained after melatonin treatment, defined as immobility of more than 1 s, and more time at the top of the tank [151]. Stress biomarkers were further investigated. Lipid peroxidation increased in the group subjected to stress compared to the control. Melatonin counteracts this response and increased catalase activity, demonstrating its antioxidant activity. Cortisol also increased after stress conditions in zebrafish, a response also inhibited by melatonin. These changes in melatonin-treated stressed zebrafish were followed by a reduction in locomotor activity, which was explained by the sleep-promoting capacity of melatonin [152].

### 3.7. Melatonin, Aging, and Chronodisruption in Zebrafish

Aging affects both the central and peripheral circadian clocks, leading to the development of age-related pathologies. Among others, aging affects rhythms including sleep, body temperature, hormone secretion, gastrointestinal, cardiovascular, and renal functions [153]. In addition, premature aging can occur due to mutations in clock genes [153].

Zebrafish are considered mature at 6 months; they can live up to 6 years and at 2 years old, they present age-dependent physiological and cognitive changes [154]. For this reason, the zebrafish emerges as a model to study the mechanisms of aging [155].

It has been observed that aged zebrafish show alterations in the circadian system, reducing activity, sleep, and melatonin production. Furthermore, there is a reduction in the expression of the clock genes Bmal1 and Per1 in addition to a phase delay of Bmal1 with age. The basal threshold of daytime activation increases with age and, therefore, the level of alertness decreases by the reduction of the circadian amplitude and/or the lack of sleep. Although melatonin production decreases with age, the expression of its receptors is not altered. Melatonin treatment in aged zebrafish promotes sleep, restores circadian rhythms and cognitive performance, and lowers the threshold for diurnal excitation.

## 4. Prospective Research

This review pretends to clarify the roles of melatonin in zebrafish physiology and pathophysiology. The zebrafish becomes an outstanding model of disease, from neurodegeneration to cardiovascular ones, and from circadian organization to psychiatric diseases. Many other diseases are being studying in either wild-type or mutant zebrafish, due to the ability to obtain transgenic fish and the knowledge of the zebrafish genome. Moreover, melatonin is rising as an excellent drug to treat a series of diseases, with almost 30 thousand publications in PubMed. Melatonin effects in humans, however, are not well-characterized. A better comprehension of its mechanism of action, analysis of its receptor-mediated, and non-receptor-mediated effects in zebrafish models of human diseases, without forgetting toxicity, is mandatory to have a basis for the clinical use of melatonin. Therefore, we encourage here melatonin-zebrafish-research to address these mechanistic features of melatonin that may give a major boost in the field for its clinical application.

## 5. Concluding Remarks—Importance to Humans

The interest of the zebrafish as an experimental model of human disease directly depends on the two main features highlighted in this review. On the one hand, there is great similarity between zebrafish and human genomes, and the fact that the zebrafish genome is sequenced, enables us to know the function of most of the genes, and the ease of zebrafish gene manipulation. The second feature is the transparency of the zebrafish embryo during development, which enables us to see pathological malformations during disease progression. These features allow us to study the phenotypic, genomic, and molecular mechanisms of diseases, and at the time to analyze the efficacy and toxicity of therapeutic drugs for human disorders. The diseases that we commented in this review affect a large part of the population. Thanks to the research in these experimental models, melatonin’s mechanisms of action have been reported, and its anti-inflammatory and antioxidant features make it an excellent candidate against a series of human diseases.

## Figures and Tables

**Figure 1 ijms-23-07438-f001:**
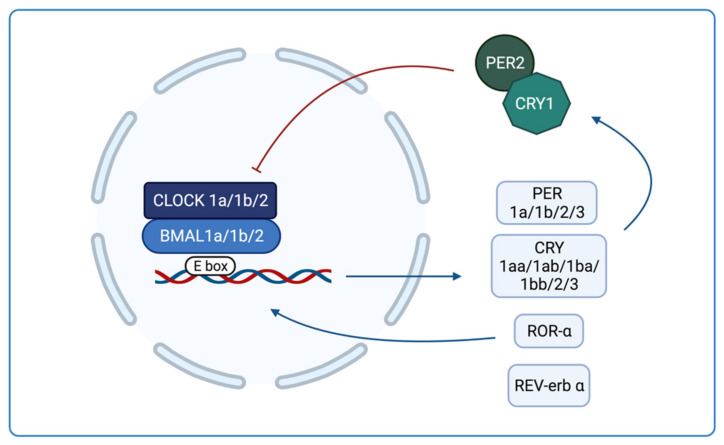
Molecular loop of zebrafish clock genes. Clock and Bmal1 form a heterodimer that activate the transcription of Per, Cry, ROR-α y REV-erb α. PER and CRY dimerize and inhibit the activity of Clock and Bmal1. ROR-α activates the BMAL1 expression and REV-erb α inhibits it.

**Figure 2 ijms-23-07438-f002:**
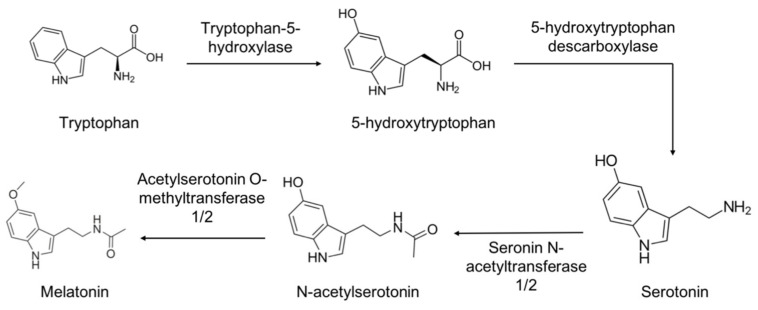
Synthesis pathway of melatonin in zebrafish. Tryptophan is uptake by the pinealocyte and hydroxylated by the enzyme tryptophan-5-hydroxylase, generating 5-hydroxytryptophan which is decarboxylated by 5-hydroxytryptophan decarboxylase yielding serotonin. Serotonin is acetylated by AANAT1 or 2, originating N-acetylserotonin. Finally, serotonin is methylated by ASMT1 or 2 producing melatonin.

**Figure 3 ijms-23-07438-f003:**
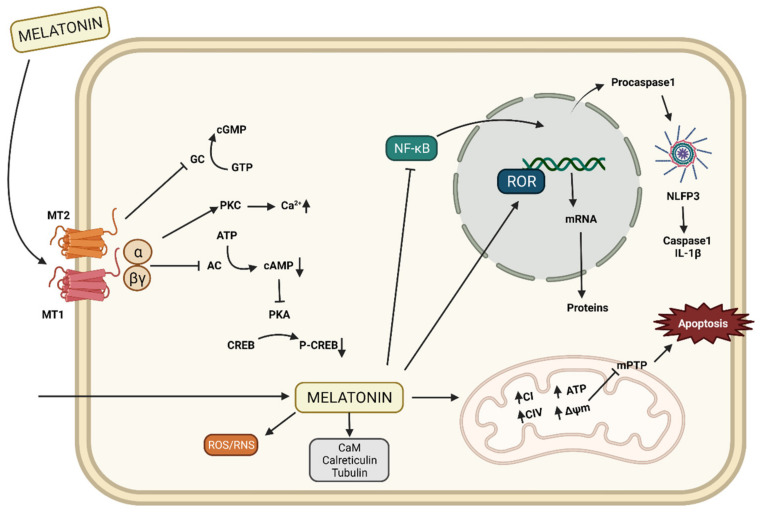
Mechanism of action of melatonin. Melatonin has 2 membrane receptors, MT1 and MT2, coupled to G proteins. Through them it inhibits cAMP and cGMP levels, CREB phosphorylation, and increases calcium levels. It also has nuclear receptors, ROR, through which it regulates gene transcription. In addition, melatonin into the cell interacts with molecules such as calmodulin, calreticulin, and tubulin, and scavenges ROS/RNS. It also inhibits the translocation of NF-kB to the nucleus where it promotes the release of procaspase 1 to the cytosol, preventing the formation of the NLRP3 inflammasome and activation of caspase 1 and IL-1β. Finally, melatonin maintains mitochondrial homeostasis, increasing the activity of the respiratory complexes I, III, and IV, ATP production, reducing membrane potential and thus closing the mTP preventing apoptosis.

## Data Availability

On request to the authors.

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
