# Peer review of "The Zebrafish, an Outstanding Model for Biomedical Research in the Field of Melatonin and Human Diseases"

_ijms, 2022, doi:10.3390/ijms23137438_

Round 1

Reviewer 1 Report

The work presented for review is a solidly prepared review of the available literature as well as a clear description of an important problem. The work is written in an interesting way and leads in a simple way to the discussed topic. Due to the limitations in performing classical studies on laboratory animals, the presented research model is interesting for research centers that in their everyday work face limitations in performing studies on traditional experimental models. The data presented in this paper convince to use this research model.

Therefore, I accept the presented work in its presented form.

Author Response

Thanks a lot for your positive comments.

Reviewer 2 Report

I suggest that the authors will dedicate a concluding paragraph emphasizing  the relevance of all the data presented to humans. 

Author Response

Thanks for the comments that positively improved the manuscript. A final paragraph suggesting the relevance of zebrafish and melatonin for human diseases.

This manuscript is a resubmission of an earlier submission. The following is a list of the peer review reports and author responses from that submission.

Round 1

Reviewer 1 Report

This manuscript examines the parallels between melatonin functions in mammals and in zebrafish to reflect on the use of zebrafish to model melatonin metabolism and function. The idea is laudable, and I am sure that such a manuscript would be of great interest. However, as is, the manuscript is poorly organized, poorly written, and lacks comparative structure. As such it is not informative.

The figures do not reflect the critical focus of this manuscript. They are simply reporting information presented elsewhere without any connection to zebrafish. At a minimum, the figures should be organized as a comparison with the zebrafish homologs and (supposed) relative mechanisms clearly indicated. E.g., how do the redundant Bmal, clock, cry function in zebrafish? As is the paper has a list of poorly-described information that does not deliver on the promise in the title for a discussion on zebrafish. Also, figure legends should clearly indicate what system(s) they represent.

The text should be logically organized. Abbreviations should be in extenso the first time they are mentioned (e.g., Line 44: dpf).

Lines 119-121” “Furthermore, it has been shown that the cell cycle is under the control of light because the cells enter the S phase of the cell cycle at the end of the day, when the larvae are exposed to the light-dark rhythm [18].” Follows a sentence on adults and it reads a non-sequitur.

Several times the text is cryptic and readers without specific knowledge would miss the point. It is an author’s duty to be clear and include novice readers. One example: Lines 59-63 imply, yet never mention the importance of zebrafish for drug screening: “In addition, the use of zebrafish is much cheaper than small mammals including mice and rats, because its small size makes the maintenance of the colony at relatively low cost, and it requires smaller volumes of chemicals and drugs to be tested, since these are administered directly into the water and are absorbed through the skin of the fish.”

Lines 408-410, unclear what the authors really want to say: “Moreover, in previous studies where mice were treated with pentylenetetrazol (PTZ), melatonin was able to reduce seizures and pinealectomized [103], whereas melatonin counteracted epileptic status in a 24-months old child [103].”

Lines 127-128: “The pineal gland and sleep regulatory mechanisms are highly conserved in zebrafish, although in a much simpler way.” Simpler than what?

Have pineal roles been demonstrated for zebrafish? Authors make it sounds speculative.

Lines 93-110: Description of the fish cry gene expression is imprecise and confusing.

Hypocretin: comes out abruptly and is not connected with the rest of the manuscript.

The section on pesticides is unlinked to the rest of the manuscript and as is, lack utility. Lines 424-426: “pesticide deltamethrin (DM) produces cardiac …. embryonic development and behavior of zebrafish [108].”

Line 453: “So, melatonin has become an option to reduce obesity.” With the information given, this conclusion is unwarranted.

Similarly, lines 483-483: “The common use of melatonin as an anxiolytic is known [121, 122] and zebrafish is an excellent model for research in pharmacology, due to its potential for the detection of central nervous system drugs.”

Description of PD is insufficient and to generic. Considering the work done by this lab one would expect to find a better source of information about the disease and how zebrafish can *specifically* model PD.

Melatonin is amphipathic and can traverse cell membranes easily. The melatonin nuclear receptors existence and function is deeply controversial and should be discussed with proper perspective and referring to the published literature at the regard. A good example of a balanced discussion was done by Ma, Molecules 2021, 26, 2693.

English must also be corrected. There are too many errors and typos to be listed here, so a few examples are reported: Lines 21-23: “Extrapineal melatonin functions include its role as an antioxidant, both inducing the endogenous antioxidants enzymes and scavenging free radicals; as an anti-inflammatory through the regulation of the NF-κB/NLRP3 inflammasome pathway, and maintains mitochondrial homeostasis.”

Lines 51-52: “This is because it offers many advantages over other vertebrate animal models: their small size, which makes them easier to manipulate”

Line 43: “all organs appear and cardiac physiology is stable at 4 dpf” seems to contradict “most of the organs are fully formed at 2 dpf”

Line 32: “vertebrate of the Cyprinidae family, corresponding to teleost and freshwater fish” Authors mean something else than “corresponding”.

Lines 140-141 “Dermatologist Aaron in 1958-59 Lerner isolated a product from the beef pineal gland”

Proper genetic notation should be respected (e.g., gene names italicized)

Lines 130-132: “In addition, these pineal photoreceptor cells contain the genes that form the intrinsic circadian clock previously described, very similar to that of retinal cells [21].” Genetic error: genes are in all cells…

Similarly: Lines 170-171: “As mentioned, the zebrafish has two aanat genes, as a result of duplications of the genome and they show different distributions. “ authors must mean gene products.

Line 356: “to hereditary mutations” this does not read properly. There are undoubtedly somatic mutations, but the context in which this text is does not warrant that line of thought. Text need clarification.

Similar, but for molecular biology: Line 199: “Membrane receptors are G protein-coupled receptors”. What about RTKs, TNFs etc?

“Zebrafish are a good model for studying Parkinson's because they have the genes whose mutations produce familial Parkinson's.” This is not warranted. Having similar genes is not enough to have equivalent function.

Too simplistic for being informative: Lines 211-212: “The mechanisms for regulating the expression of these membrane receptors are very complex, depending on factors such as location, the light/dark cycle, the phase of the circadian rhythm, among others [44].”

Citation needed: Line 238-243: “In our group, we have observed the independent effects of membrane receptors in a zebrafish model of Parkinson's disease by blocking the melatonin receptors Mel1a and …. treatment to a model of MPTP-induced Parkinson's disease in zebrafish.”

Also needing citation (not sentences later) Line 386: “Melatonin has been shown to counteract the reduction in autophagy and the increase in mitochondrial fusion induced by MPTP.”

Author Response

This manuscript examines the parallels between melatonin functions in mammals and in zebrafish to reflect on the use of zebrafish to model melatonin metabolism and function. The idea is laudable, and I am sure that such a manuscript would be of great interest. However, as is, the manuscript is poorly organized, poorly written, and lacks comparative structure. As such it is not informative.

Thanks you for your comments. We addressed all of them accordingly.

The figures do not reflect the critical focus of this manuscript. They are simply reporting information presented elsewhere without any connection to zebrafish. At a minimum, the figures should be organized as a comparison with the zebrafish homologs and (supposed) relative mechanisms clearly indicated. E.g., how do the redundant Bmal, clock, cry function in zebrafish? As is the paper has a list of poorly-described information that does not deliver on the promise in the title for a discussion on zebrafish. Also, figure legends should clearly indicate what system(s) they represent.

Figures were redrawn and adapted to zebrafish physiology.

The text should be logically organized. Abbreviations should be in extenso the first time they are mentioned (e.g., Line 44: dpf).

Corrected.

Lines 119-121” “Furthermore, it has been shown that the cell cycle is under the control of light because the cells enter the S phase of the cell cycle at the end of the day, when the larvae are exposed to the light-dark rhythm [18].” Follows a sentence on adults and it reads a non-sequitur.

Paragraph revised and corrected.

Several times the text is cryptic and readers without specific knowledge would miss the point. It is an author’s duty to be clear and include novice readers. One example: Lines 59-63 imply, yet never mention the importance of zebrafish for drug screening: “In addition, the use of zebrafish is much cheaper than small mammals including mice and rats, because its small size makes the maintenance of the colony at relatively low cost, and it requires smaller volumes of chemicals and drugs to be tested, since these are administered directly into the water and are absorbed through the skin of the fish.”

Information revised.

Lines 408-410, unclear what the authors really want to say: “Moreover, in previous studies where mice were treated with pentylenetetrazol (PTZ), melatonin was able to reduce seizures and pinealectomized [103], whereas melatonin counteracted epileptic status in a 24-months old child [103].”

Sentence removed.

Lines 127-128: “The pineal gland and sleep regulatory mechanisms are highly conserved in zebrafish, although in a much simpler way.” Simpler than what?

Sentence revised.

Have pineal roles been demonstrated for zebrafish? Authors make it sounds speculative.

Information was added to this point.

Lines 93-110: Description of the fish cry gene expression is imprecise and confusing.

revised and updated.

Hypocretin: comes out abruptly and is not connected with the rest of the manuscript.

I believe that hypocretin is a new partner in the circadian control of zebrafish, and this is clarified.

The section on pesticides is unlinked to the rest of the manuscript and as is, lack utility. Lines 424-426: “pesticide deltamethrin (DM) produces cardiac …. embryonic development and behavior of zebrafish [108].”

information clarified.

Line 453: “So, melatonin has become an option to reduce obesity.” With the information given, this conclusion is unwarranted.

Sentence removed.

Similarly, lines 483-483: “The common use of melatonin as an anxiolytic is known [121, 122] and zebrafish is an excellent model for research in pharmacology, due to its potential for the detection of central nervous system drugs.”

Studies conducted to analyze the anxiolytic effect of melatonin in zebrafish has been conducted, and they are now included in the manuscript.

Description of PD is insufficient and to generic. Considering the work done by this lab one would expect to find a better source of information about the disease and how zebrafish can *specifically* model PD.

Because the review is focused on different pathological conditions and not only in PD, we decide to avoid an excess of information of PD only and to maintain an equilibrium in the information.

Melatonin is amphipathic and can traverse cell membranes easily. The melatonin nuclear receptors existence and function is deeply controversial and should be discussed with proper perspective and referring to the published literature at the regard. A good example of a balanced discussion was done by Ma, Molecules 2021, 26, 2693.

Melatonin does not traverse cell membranes easily; we demonstrated this feature in several tissues of rats (Venegas C, García JA, Escames G, Ortiz F, López A, Doerrier C, García Corzo L, López LC, Reiter RJ , Acuña-Castroviejo D. "Extrapineal melatonin: Analysis of its subcellular distribution and daily fluctuations." J Pineal Res;2012 52:217-227). Nuclear melatonin receptors have been also supported by many experimental evidences. In zebrafish, however, the existence and roles of nuclear receptors of melatonin in zebrafish is yet unknown.

English must also be corrected. There are too many errors and typos to be listed here, so a few examples are reported: Lines 21-23: “Extrapineal melatonin functions include its role as an antioxidant, both inducing the endogenous antioxidants enzymes and scavenging free radicals; as an anti-inflammatory through the regulation of the NF-κB/NLRP3 inflammasome pathway, and maintains mitochondrial homeostasis.”

Corrected.

Lines 51-52: “This is because it offers many advantages over other vertebrate animal models: their small size, which makes them easier to manipulate”

Line 43: “all organs appear and cardiac physiology is stable at 4 dpf” seems to contradict “most of the organs are fully formed at 2 dpf”

Line 32: “vertebrate of the Cyprinidae family, corresponding to teleost and freshwater fish” Authors mean something else than “corresponding”.

Lines 140-141 “Dermatologist Aaron in 1958-59 Lerner isolated a product from the beef pineal gland”

 English revised.

Proper genetic notation should be respected (e.g., gene names italicized)

Corrected.

Lines 130-132: “In addition, these pineal photoreceptor cells contain the genes that form the intrinsic circadian clock previously described, very similar to that of retinal cells [21].” Genetic error: genes are in all cells…

 Corrected.

Similarly: Lines 170-171: “As mentioned, the zebrafish has two aanat genes, as a result of duplications of the genome and they show different distributions. “ authors must mean gene products.

Corrected.

Line 356: “to hereditary mutations” this does not read properly. There are undoubtedly somatic mutations, but the context in which this text is does not warrant that line of thought. Text need clarification.

Corrected.

Similar, but for molecular biology: Line 199: “Membrane receptors are G protein-coupled receptors”. What about RTKs, TNFs etc?

Corrected.

“Zebrafish are a good model for studying Parkinson's because they have the genes whose mutations produce familial Parkinson's.” This is not warranted. Having similar genes is not enough to have equivalent function.

 Corrected.

Too simplistic for being informative: Lines 211-212: “The mechanisms for regulating the expression of these membrane receptors are very complex, depending on factors such as location, the light/dark cycle, the phase of the circadian rhythm, among others [44].”

 Revised.

Citation needed: Line 238-243: “In our group, we have observed the independent effects of membrane receptors in a zebrafish model of Parkinson's disease by blocking the melatonin receptors Mel1a and …. treatment to a model of MPTP-induced Parkinson's disease in zebrafish.”

 OK.

Also needing citation (not sentences later) Line 386: “Melatonin has been shown to counteract the reduction in autophagy and the increase in mitochondrial fusion induced by MPTP.”

OK.

Reviewer 2 Report

The author described the physiology and production of melatonin as well as role of melatonin in different aspects in Zebrafish model. i fund the paper interesting and well structured. however I believe some work need to be done before publication.  

First of all, there is lots of statements which need citation. for instance first two paragraphs, L151-158, L474-481 needs more citation.

L129-130 please re write.

L 70 I personally disagree. Zebrafish has quite many physiological dissimilarities in embryo development with humans obviously one is ex utero and the other one is in utero

Role of melatonin in reproduction and thermoregulation is discussed very briefly or not discussed at all. 

Article lack prospective research

Author Response

The author described the physiology and production of melatonin as well as role of melatonin in different aspects in Zebrafish model. i fund the paper interesting and well structured. however I believe some work need to be done before publication.  

First of all, there is lots of statements which need citation. for instance first two paragraphs, L151-158, L474-481 needs more citation.

Revised.

L129-130 please re write.

Revised.

L 70 I personally disagree. Zebrafish has quite many physiological dissimilarities in embryo development with humans obviously one is ex utero and the other one is in utero

I agree with this observation but, besides of this, other developmental procedures are quite similar.

Role of melatonin in reproduction and thermoregulation is discussed very briefly or not discussed at all. 

There is not much information regarding melatonin and thermoregulation in zebrafish, and some of it is related to the role of temperature changers in clock genes expression. A reference of this effect is included in page 3.

Article lack prospective research

We added a brief analysis of this point at the end of the discussion.

Round 2

Reviewer 2 Report

The authors improved the MS and added future prospective, although changes were not done in track change mode.